# Predictive and Prognostic Factors in HCC Patients Treated with Sorafenib

**DOI:** 10.3390/medicina55100707

**Published:** 2019-10-21

**Authors:** Oronzo Brunetti, Antonio Gnoni, Antonella Licchetta, Vito Longo, Angela Calabrese, Antonella Argentiero, Sabina Delcuratolo, Antonio Giovanni Solimando, Andrea Casadei-Gardini, Nicola Silvestris

**Affiliations:** 1Medical Oncology Unit, National Cancer Research Centre, IRCCS Istituto Tumori “Giovanni Paolo II”, 70124 Bari, Italy; argentieroantonella@gmail.com (A.A.); antoniogiovannisolimando@gmail.com (A.G.S.); n.silvestris@oncologico.bari.it (N.S.); 2Medical Oncology Unit, “S. Cuore di Gesù” Hospital, 73014 Gallipoli, Italy; drgnoni.antonio@libero.it (A.G.); antonellalicchetta@libero.it (A.L.); 3Medical Thoracic Oncology Unit, IRCCS Istituto Tumori “Giovanni Paolo II”, 70124 Bari, Italy; vito.longo79@tiscali.it; 4Radiology Unit, National Cancer Research Centre, IRCCS Istituto Tumori “Giovanni Paolo II”, 70124 Bari, Italy; acalabrese22@gmail.com; 5Scientific Direction, National Cancer Research Centre, IRCCS Istituto Tumori “Giovanni Paolo II”, 70124 Bari, Italy; delcuratolo.sa@gmail.com; 6Department of Biomedical Sciences and Human Oncology, Section of Internal Medicine “G. Baccelli”, University of Bari Medical School, 70124 Bari, Italy; 7Department of Medical Oncology, IstitutoScientifico Romagnolo per Lo Studio e Cura Dei Tumori (IRST) IRCCS, 47014 Meldola, Italy; casadeigardini@gmail.com; 8Department of Oncology and Haematology, University Hospital of Modena, 41125 Modena, Italy; 9Department of Biomedical Sciences and Human Oncology, University of Bari “Aldo Moro”, 70124 Bari, Italy

**Keywords:** Sorafenib, hepatocellular carcinoma, prognostic factors, predictive factors

## Abstract

Sorafenib is an oral kinase inhibitor that enhances survival in patients affected by advanced hepatocellular carcinoma (HCC). According to the results of two registrative trials, this drug represents a gold quality standard in the first line treatment of advanced HCC. Recently, lenvatinib showed similar results in terms of survival in a non-inferiority randomized trial study considering the same subset of patients. Unlike other targeted therapies, predictive and prognostic markers in HCC patients treated with sorafenib are lacking. Their identification could help clinicians in the daily management of these patients, mostly in light of the new therapeutic options available in the first.

## 1. Introduction

Sorafenib (NEXAVAR^®^) is a small molecule classified as an oral multi-targeted tyrosine kinase inhibitor (TKI) since it inhibits platelet-derived growth factor receptors (PDGFR), vascular endothelial growth factor (VEGF-R), tyrosine-protein kinase KIT, and fibroblast growth factor receptors FGFR1. As is known, this TKI impairs angiogenesis, cancer proliferation, and cell apoptosis (Figure 1) [1].

In 2007, the FDA approved this drug in unresectable and advanced HCC, according to the results of Sorafenib Hepatocellular Carcinoma Assessment Randomized Protocol (SHARP) [2] and Asia-Pacific [3] randomized phase III studies. In the SHARP trial, median overall survival (mOS) was 10.7 months and 7.9 months for sorafenib and placebo groups, respectively (hazard ratio (HR) 0.69,95% confidence interval (CI) 0.55 to 0.87, *p* < 0.001) [2]. In the Asia-Pacific trial as well, sorafenib-treated patients showed an mOS significantly higher as a result compared to the placebo group (6.5 months vs. 4.2 months, respectively, HR 0.68, 95% CI 0.50–0.93, *p* = 0.014) [3].

Hand-foot skin reaction (HFSR), fatigue, diarrhea, anorexia, and weight reduction were the most common treatment-related adverse effects (AE) in both studies [2,3].

Several randomized clinical trials were performed subsequently and evaluated other tirosine kinase inhibitors (TKIs) compared with sorafenib, which did not report an improvement in terms of clinical outcomes [4,5,6]. In particular, sunitinib [4] and linifanib [5] did not achieve superior mOS compared to sorafenib (8.1 months vs. 10.0 months, respectively, *p* = 0.0019–9.1 months vs. 9.8 months, HR 1.046,95% CI, 0.896 to 1.221, respectively). Furthermore, sorafenib achieved a significant improvement in terms of mOS compared to brivanib (9.9 months vs. 9.5 months *p* > 0.05) [6].

Methodological bias influenced these negative results. In particular, the bias includes the lack of phase II studies evaluating liver toxicity (required in cirrhotic patients with HCC), the need for stronger secondary endpoints (i.e., time to progression, TTP, and objective response rate, ORR), according to the modified RECIST (mRECIST) criteria, and the lack of predictive biomarkers [7].

Only lenvatinib showed similar results in terms of survival in a non-inferiority randomized trial, with an mOS for lenvatinib of 13.6 months (95% CI 12.1–14.9) compared to sorafenib (12.3 months, 95% CI 10.4–13.9, HR 0.92) with different toxicity profiles [8].

In this manuscript, we reviewed the data available in the literature with the aim to try to answer the following question. Is it possible today to select patients as candidates for sorafenib according to clinical or biological predictive and/or prognostic markers?

## 2. Clinical Predictive/Prognostic Markers

### 2.1. Barcelona Clinic Liver Cancer Staging and Child-Pugh Cirrhosis Classifications

Although it is an expected situation, Barcelona Clinic Liver Cancer (BCLC) staging and Child-Pugh (CP) cirrhosis classifications resemble the most important criteria in the selection of HCC patients who are suitable for treatment with sorafenib. In the registrative trials [2,3], almost all enrolled patients were CP-A. Correlation between CP and the response to sorafenib has been confirmed in two prospective studies. The first considered 120 patients [9] with mOS of 13 months and 4.5 months, respectively (*p* = 0.0008). The second evaluated 300 patients [10] with mOSof 10.0 months and 3.8 months, respectively (*p* < 0.001). The GIDEON trial [11], which is an observational registry study, evaluated the survival and safety of sorafenib in 1968 and 666 patients with a CP-A and CP-B status, respectively. The mOSwere 13.6 months (95% CI 12.8–14.7) and 5.2 months (95% CI 4.6–6.3) in CP-A and CP-B patients, respectively.

In addition, BCLC has been evaluated as the clinical predictive criteria for the response to sorafenib. In the SHARP trial, BCLC B and BCLC C achieved mOS of 14.5 months and 9.7 months, respectively [2]. Later, the Italian study SOFIA [12] compared mOS of BCLC B and BCLC C patients treated with sorafenib, and found a significant advantage for BCLC B (mOS: 20.6 months and 8.4 months, *p* < 0.0001, respectively). More recently, a pooled analysis of SHARP and Asia-Pacific trials [13] endorsed previous results, which demonstrated that BCLC B had a better survival than those with BCLC C HCC patients (HR = 1.59, *p* = 0.02). This confirms the predictive role of BCLC staging.

### 2.2. Viral Status

Since hepatitis B virus (HBV) and hepatitis C virus (HCV) represent the main causes of HCC, the viral status has been analyzed in several studies. Data from the SHARP and the Asia-pacific trials pooled in the analysis performed by Bruix et al. [13] demonstrated that non-HCV related HCC has a worse OS (HR = 0.7, *p* = 0.02), while HBV infection did not achieve a significant difference in patients treated with sorafenib (HR = 1.128, *p* = 0.4538) compared to HBV-positive HCC patients. In a more recent meta-analysis of sorafenib and lenvatinib trials, lenvatinib was shown to be the best agent for both HBV and HCV infected patients, which presented a more favorable HR versus sorafenib treated with HCC (HR 0.83, 95% CI 0.68–1.01 and HR 0.91, 95%CI 0.66–1.25, respectively) [14].

### 2.3. Diabetes and Use of Oral Antidiabetics

Recently, diabetes and use of oral antidiabetics have been analyzed in HCC patients. Diabetes mellitus is as a risk factor in the development of HCC mostly in patients who are not HBV/HCV positive [15,16]. On the contrary, Di Costanzo et al., in an observational study, did not confirm the prognostic role of diabetes with mOS of 9 months and 10 months in HCC non-diabetic and diabetic patients, respectively (*p* = 0.535). Furthermore, median time to progression (mTTP) was longer in diabetic patients (*p* = 0.038). As for oral antidiabetics, the role of metformin is still uncertain. In a propensity score-matched cohort analysis, the combination of metformin and radiation therapy in unresectable HCC prolonged the OS rate (two-year,76% vs. 37%, *p* = 0.022) [17]. On the contrary, in a retrospective study [18], metformin reduced sorafenib activity in HCC patients with type II diabetes mellitus with median progression free survival (mPFS) of 2.6 months and 5.0 months and mOS of 10.4 months and 15.1 months for patients chronically-treated with or without metformin, respectively. These data were validated in a case-series of more than 279 HCC patients [19]. There are no clinical trials on the use of metformin in HCC patients. So far, the efficacy of metformin in diabeticand non-diabetic patients is still unknown.

### 2.4. Adverse Events Due to Sorafenib

Among clinical predictive markers, several studies displayed a positive correlation in HCC patients between survival and adverse events due to sorafenib. The mOS of 634 HCC patients who presented any grade of toxicities related to sorafenib (Hand-Foot Skin Reaction - HFSR, hypertension, diarrhea) and mOS significantly improved when compared to patients without adverse events (8.8 months vs. 5.4 months, respectively, IQR 2.7–8.8, log-rank *p* = 0.004) [20].

Reig et al. assessed 147 patients treated with sorafenib [21]. They observed that HFSR represented an independent predictive factor of better survival, since patients with early HFSR displayed better OS than the patients who did not show this adverse event within the first 60 days of treatment (18.2 months vs. 10.1 months, respectively, *p* = 0.009) [21]. A recent metanalysis of 12 cohort studies including 1017 patients confirmed the significant correlation between HFSR and the response to sorafenib (pooled HR for mOS of 0.45,95% CI 0.36, 0.55, *p* < 0.00001, I^2^ = 35%) and time to progression (TTP) of 0.41 (95% CI 0.28, 0.60, *p* < 0.00001, I^2^ = 0%) [22].

The predictive role of hypertension appears uncertain. A study involving 61 HCC patients demonstrated that those with hypertension who developed this side effect 15 days after beginning sorafenib compared to others who had better mPFS (6.0 months vs. 2.5 months, *p* < 0.001) and mOS (14.6 months vs. 3.9 months; *p* = 0.003). On the contrary, in the studies by Shin SY et al. [23] and Otsuka T et al. [24], hypertension was not related to OS (*p* = 0.262 and *p* = 0.332, respectively).

Regarding diarrhea, Koschny R et al. [25] demonstrated a significant correlation between the grade of this symptom and mOS (grade 2–3 vs. 0–1: 11.8 months vs. 4.2 months—95% CI 6.9–16.6 vs. 95% CI 0.0–9.1, respectively, *p* = 0.009).

In a large, multi-centric retrospective analysis, Di Costanzo et al. [26] demonstrated that mOS was 14.4 months (95% CI 12.0–16.8) and 5.8 months (95% CI 4.6–7.1) in patients with and without HFSR (*p* = 0.005). Furthermore, patients with and without hypertension achieved mOS at 15.1 months (95% CI 12.9–17.3) and 7.5 months (95% CI 5.9–9.2) (both *p* < 0.001), respectively. The mOSwere 15.6 months (95% CI 11.1–20.1) and 9.2 months (95% CI 7.6–10.8) in patients with and without diarrhea, respectively (both *p* < 0.001). When a score from 0 to 3 was assigned depending on the number of side effects suffered, a progressive increase in survival was noted. In particular, mOS was 7.9 months, 9.2 months, 15.1 months, and 23.9 months in patients with scores of 0, 1, 2, and 3, respectively (*p* < 0.001).

It would be interesting to discover if the disease background, family history, and treatment history could be considered prognostic biomarkers in the sorafenib response. To date, no data are available in literature concerning these topics.

## 3. Biological Predictive Markers

In the era of target therapy and liquid dynamic medicine [27], plasmatic and histological biomarkers have always been used as predictive markers of assumed responses to target therapy. This holds true in HCC patients treated with sorafenib (Figure 2) as well.

### 3.1. Alpha-Fetoprotein

High alpha-fetoprotein (AFP)values are observed in about half of HCC patients. So far, it is still the principal serological biomarker used for managing this malignancy, even if it is elevated in cirrhotic patients as well [28]. In the SHARP trial, alpha fetoprotein (AFP) plasma levels >200 ng/mL were a negative prognostic marker [2]. These data have been recently confirmed in a pooled analysis of the two registrative trials [13]. Furthermore, an early decrease of AFP seems to be a predictive biomarker [29,30]. Shao et al. [29] defined patients with a reduction of more than 20% from baseline serum levels after two to four weeks of treatment as early AFP responders. Responders were compared with non-responders with a significantly improved overall response rate (ORR) (33% vs. 8%; *p* = 0.037) and disease control rate (DCR) (83% vs. 35%, *p* = 0.002), respectively. Moreover, mPFSs were 7.5 months vs. 1.9 months (*p* = 0.001) and mOS15.3 months vs. 4.1 months (*p* = 0.019) for responding and non-responding patients, respectively. Sanchez et al. [30] demonstrated that a decrease of more than 20% AFP at 6–8 weeks from baseline was a positive predictive marker of response to sorafenib. In a multivariate analysis (*p* = 0.002), with mOS of 18 months and 10 months (*p* = 0.004) for responding and non-responding patients, respectively. Furthermore, Nakazawa et al. [31] defined as an increase in AFP when its serum levels were 20% more than the baseline. An early increase of AFP after sorafenib was a significant negative predictive factor, since mOS (*p* < 0.001, HR 4.14; 95% CI 1.946–8.811) and mPFS (*p* = 0.001, HR 2.852, 95% CI 1.524–5.337) of these patients were worse than the others.

### 3.2. Angiogenetic Markers

Angiogenetic markers were analyzed in several studies, since angiogenesis represents one of the most activated pathways in HCC [32]. Among angiogenetic factors, the most studied have been angiopoietin-2 (Ang-2) and vascular endothelial growth factor-A (VEGF-A). In the SHARP study [2], baseline VEGF and ANG2 plasma levels were prognostic factors in sorafenib and placebo-treated HCC. Anyway, none of them led to a predictive biomarker of sorafenib.

Tsuchiya et al. [33] revealed that a decrease of plasma VEGF concentrations with sorafenib treatment after eight weeks was a predictor of better mOS than others (30.9 months vs. 14.4 months, *p* = 0.038).

In a mouse model of HCC, Horwitz et al. demonstrated that VEGF-A gene amplification was related to better survival when compared to non-amplified tumors [34]. Moreover, they verified the data on HCC serum of patients treated with sorafenib in vivo. So far, the mOS were10 months and not achieved for patients with negative (47 patients) and positive (7 patients) VEGF-A gene amplification, respectively (*p* = 0.029).

In a recent study [35], the circulating cell-free DNA (cfDNA) concentrations of VEGF were analyzed in HCC patients treated with sorafenib. Patients whose disease progressed with sorafenib had significantly higher cfDNA levels than the others (0.82 ng/μLvs.0.63 ng/μL, *p* = 0.006). Moreover, when patients were classified into cfDNA-high-low groups (above and below the median of cfDNA concentrations of VEGF, respectively), a significantly worse TTP (2.2 months vs. 4.1 months, respectively, HR = 1.71, *p* = 0.002) and OS (4.1 months vs. 14.8 months, respectively, HR = 3.50, *p* < 0.0001) were achieved in the first group than in the latter.

In addition, Single Nucleotide Polymorphisms (SNPs)of VEGF were analyzed. In a study by Scartozzi et al. [36], aunivariate analysis of VEGF-A alleles C of rs25648, T of rs833061, C of rs699947, C of rs2010963, VEGF-C alleles T of rs4604006, G of rs664393, VEGFR-2 alleles C of rs2071559, C of rs2305948 showed significant predictive factors of PFS and OS in sorafenib-treated HCC. In the multivariate analysis, VEGF-A rs2010963 and VEGF-C rs4604006 were independent factors influencing PFS (HR = 0.25, 95% CI: 0.19–1.02, *p* = 0.0376 and HR = 0.22, 95% CI: 0.14–0.81, *p* = 0.004, respectively) and OS (HR = 0.28, 95% CI: 0.23–0.96, *p* = 0.02 and HR = 0.25, 95% CI: 0.17–0.99, *p* = 0.04, respectively).

Miyahara et al. [37] described a negative predictive outcome in HCC patients with high Ang-2 serum levels before sorafenib (HR = 2.51, 95% CI: 1.01–6.57, *p* = 0.048). More recently, the expression of a SNP for ANGPT2, an Ang2 gene, and a rs55633437 GG genotype showed a significantly longer PFS (*p* < 0.001) and OS (*p* < 0.001) than those with the other genotypes (GT+TT) [38]. In any case, even if these results describe a potential prognostic role of Ang-2 or its polymorphisms in HCC, its role in predicting a response to sorafenib should be verified.

In another interesting study on the angiogenic gene [39], eNOS polymorphisms were analyzed in relation to PFS and OS. In univariate and multivariate analyses, a training cohort of HCC patients homozygous for endothelial nitric oxide synthase (eNOS) haplotype (HT1:T-4b at eNOS-786/eNOS VNTR) had a worse mPFS (2.6 months vs. 5.8 months, HR = 5.43, 95% CI: 2.46–11.98, *p* < 0.0001) and OS (3.2 months vs. 14.6 months, HR = 2.35, 95% CI: 1.12–4.91, *p* = 0.024) when compared with other haplotypes.

Other studies are evaluating more than one anti-angiogenic biomarker. ALICE-2 study [40] evaluated the role of hypoxia-inducible factor 1-alpha (HIF-1α) and SNPs of HIF-1α, VEGF, and Ang2. The multivariate analysis demonstrated that rs12434438 (SNP of HIF-1α), rs2010963 (SNP of VEGF-A), and rs4604006 (SNP of VEGF-C) were independent factors and were predictive biomarkers of the sorafenib response. Currently, a prospective ongoing study (INNOVATE) has the aim to confirm the role of SNPs of VEGF, HIF-1α, Ang-2, and eNOS SNPs in relation to treatment with sorafenib [41].

### 3.3. Inflammatory Cells, Proteins, and Index

It must be said that the systemic inflammatory micro-environment has a strong correlation with angiogenesis, tumor invasion, and metastasis through an upregulation of inflammatory cells and cytokines (i.e., the activation of mechanisms of immune-tolerance in gastrointestinal cancer, including HCC) [42,43,44].In particular, an inflammatory micro-environment and circulating immune cells and cytokines play a significant role in HCC prognosis [45,46,47]. So far, Hu B et al. [45] used a systemic immune-inflammation (SII) index with an aim to predict the prognosis of patients after curative resection. This index was based on lymphocyte, neutrophil, and platelet counts and was able to predict survival and recurrence in HCC. Univariate and multivariate analyses revealed that the SII index was an independent predictive factor for mOS and was a prognostic factor for patients with negative AFP levels and BCLC 0/A. Afterward, Lue et al. [46] demonstrated that a neutrophil-lymphocyte ratio (NRL) ≥ 2.3 was a negative predictive biomarker in the response to sorafenib in in both univariate and multivariate analyses (*p* = 0.005 and HR 1.72, 95% CI: 1.03–2.71, respectively) in HCC European patients. More recently, similar data have been achieved in an Asiatic cohort [47]. A meta-analysis conducted on 6318 patients [48] observed that a high NLR before any treatment was predictive of a short mOS (HR: 1.54, 95% CI: 1.34 to 1.76, *p* < 0.001). In this study, authors analyzed the platelet-lymphocyte ratio (PLR) demonstrating that the increase of PLR predicted an unfavorable outcome in terms of mOS as well (HR: 1.63, 95% CI: 1.34 to 1.98, *p*<0.001). In any case, these data were not found when they were analyzed in the subgroup of sorafenib-treated patients. Casadei-Gardini et al. [49] considered SII, NLR, and PLR in a retrospective multi-center case series. They observed that patients treated with sorafenib and with SII ≥ 360 showed poorer survival outcomes when compared to patients with SII < 360 in terms of mPFS (2.6 months vs. 3.9 months, respectively, *p* < 0.026) and mOS (5.6 months vs. 13.9 months, respectively, *p* = 0.027). Patients with NLR ≥ 3 compared with those with NLR < 3, had a lower mPFS (2.6 months vs. 3.3 months, *p* < 0.049). However, no significant data were reported in terms of mOS (5.6 months vs. 13.9 months, *p* = 0.062). So far, SII and NLR could represent predictive factors for patients with advanced HCC treated with sorafenib.

Oxidative stress is a key pathogenic event in the development and progression of HCC [50]. Nuclear erythroid 2-related factor 2 (Nrf2) is a cytosolic transcription factor regulating the cellular protection by inducing anti-inflammatory, antioxidant, and cyto-protective gene expression [51]. In particular, its dysregulation enhances the resistance of cancer cells against drugs. So far, the identification of molecules targeting Nrf2 might open a new scenario for preventing and treating HCC. Preliminary data from in vitro studies showed the ability of Sorafenib to reverse 5-fluorouracil (5-FU) resistance likely through the suppression of Nrf2 expression induced by 5-FU [52], with a putative role in the Sorafenib-response prediction.

Another potential predictive factor is Insulin-like Growth Factor (IGF)-1. Eighty-three patients with high (i.e., levels ≥ the median level) baseline IGF-1 levels achieved a significantly higher disease control rate (DCR) when treated with antiangiogenic therapies (including sorafenib) than those with low levels (71% vs. 39%, respectively—*p* = 0.003) [53]. Moreover, patients with high IGF-1 levels, when compared with those with low levels, showed longer mPFS (4.3 months vs. 1.9 months, respectively—*p* = 0.014) and mOS (10.7 months vs. 3.9 months, respectively—*p* = 0.009). Multivariate analysis demonstrated that high baseline IGF-1 levels were an independent predictive factor of anti-angiogenetic drugs in terms of PFS and OS.

### 3.4. Growth Factors and Other Targets

Arao et al. analyzed a comparative genomic hybridization in frozen HCC samples from patients responsive to sorafenib [54]. Fibroblast growth factor (FGF) 3/FGF4 amplification was observed in 30% of HCC samples while it was not seen in 38 non-responsive patients (*p* = 0.006). These data were confirmed in vitro with a growth inhibitory assay, since FGF3/FGF4-amplified HCC cell lines exhibited hypersensitivity to sorafenib. To assess a complete panel of genes predictive of the sorafenib response, DNA and RNA sequencing using a fine-needle biopsy was performed in 46 patients [55]. Comparisons were conducted between the transforming growth factor (TGF) gene expression levels of progressive disease (PD)-patients and non PD-patients (74.1 vs. 20.3 median read number, respectively *p* = 0.0180) and the Platelet And Endothelial Cell Adhesion Molecule 1 (PECAM1) gene expression levels of these two groups of patients (110.2 vs. 13.2 median read number, respectively *p* = 0.0131). Both TGFa and PECAM1 gene expression levels were significantly increased in the non-PD group. Moreover, mPFS of patients with high and low neuregulin 1 (NRG1) expressions were 80 days and 90 days in sorafenib responding patients, respectively (*p* = 0.0497). So far, high TGFa and PECAM1 and low NRF1 gene levels should be predictors of response to sorefenib.

Although the B-type Raf kinase (BRAF) mutation could play a role in the response to sorafenib. BRAF, which is a protein located downstream of the Kirsten RAt Sarcoma virus (KRAS) pathway, is implicated in the response to anti-EGFR treatment [56]. In a case series of advanced gastrointestinal stromal tumor (GIST) [57], sorafenib was administrated to patients resistant to imatinib, sunitinib, and regorafenib. In these patients, BRAF was tested for mutations. So far, two BRAF wild-type patients achieved long-term disease control (49 months and 19 months, respectively), while sorafenib-resistant patients carried a BRAF V600E mutation. In a report by Casadei-Gardini et al. [58], a patient with synchronous lung cancer (LC) and HCC, treated with sorafenib, achieved a response in LC but not in liver cancer. The mutational analysis revealed a BRAF exon11 mutation (G469V) only in LC. Authors hypothesized that this mutation could be responsible for HCC resistant to sorafenib, which sheds light on a possible negative prognostic role of this mutation.

### 3.5. MiRNAs

Recently, micro RNAs (miRNAs)achieved a key role in gastrointestinal cancers [59,60,61]. In particular, up/down-regulation of several miRNAs has been reported to be able to impair the TKI response, which affects the expression of genes involved in several pathways [61,62,63]. For example, miRNA-21 could enhance resistance to sorafenib in vitro through the PTEN/Akt pathway by inhibiting autophagy [62]. MicroRNA-122 obtained sorafenib resistance to HCC cell lines through the RAS/RAF/ERK pathway [63]. Moreover, in an animal HCC model, elevated miR-122 levels were associated with a stem-like phenotype in HCC [64] associated with resistance to sorafenib. So far, an anti-miRNA122 transfection increased cell viability in sorafenib-treated HCC cells, which restored sorafenib activity HCCs. The predictive role of circulating miRNAs has also been investigated. The miRNA181a-5p levels resulted in the unique independent factor for sorefenib-treated patients achieving a DCR in 53 patients (HR 0.139, 95% CI 0.011–0.658, *p* = 0.0092) [65]. Furthermore, miR-181a-5p resulted in the only independent factor in terms of OS in multivariate analysis (HR 0.267, 95% CI 0.070–0.818, *p* = 0.0194) [65]. Sorafenib upregulated MiRNA423-5p both in vitro and in vivo and its increase from baseline to evaluation at six months correlated with the response. In fact, 75% of patients with an miR423-5p level increase achieved disease control [66]. In addition, MiR-126-3p was down-regulated after sorafenib treatment in HCC cells lines [67]. So far, Faranda et al. determined the expression levels of miR-126-3p in HCC tissues and plasma. This miRNA was down-regulated in HCC tissues compared to levels of peritumoral tissues (HCC average = 3.91 ± 0.48 vs. RQPT average = 5.84 ± 0.51, *p*-value = 0.0074). Moreover, circulating miR-126-3p expression levels were significantly higher in HCC patients when compared to control subjects (26.7 vs. 26.6 mean expression levels, *p*-value = 0.0002) [67]. In vitro data and in vivo determination led authors to hypothesize that a reduction of this miRNA could be predictive of a response to sorafenib.

In an exploratory study [68], several miRNAs (miRNA10b-3p, miRNA18a, miRNA139-5p, miRNA21, miRNA224, miRNA221) were evaluated as predictive markers for the response to sorafenib. Only miRNA10b-3p expression levels were significantly higher (fold increase = 5.8) in the subgroup of HCC patients with worse OS (*p* = 0.008) and with a putative prediction of short survival of sorafenib-treated patients.

The predictive role of miRNA has been evaluated in HCC tissue in clinical studies. In particular, high levels of miRNA-224 in HCC samples were correlated with an increase of PFS (HR = 0.28, 95% CI: 0.09–0.92, *p* = 0.029) and OS (HR = 0.0.24, 95% CI: 0.07–0.79, *p* = 0.012) in patients treated with sorafenib [69]. In another study, patients with high levels of miR-425-3p in HCC tissue treated with sorafenib achieved a better PFS (HR = 0.5, 95% CI: 0.3–0.9, *p* = 0.007) and TTP (HR = 0.4, 95% CI: 0.2–0.7, *p* = 0.0008) [70].

## 4. Conclusions

Currently, even if sorafenib still remains a gold quality standard in the first line treatment of advanced HCC patients, new targeted therapies and immunotherapies have been approved and will be approved soon. So far, several clinical and biological biomarkers have been evaluated with the aim to improve the choice of patients suitable for treatment with these drugs (Table A1). Nonetheless, we are still far from obtaining a panel useful for clinical practice. The efforts must be to identify a score [27,71], which is able to include various variables useful for perfecting the therapeutic choice in HCC.

Furthermore, in vitro and in vivo preclinical models evaluated the combination of Sorafenib with the drugs that are able to overcome its acquired resistance mechanisms [72]. Contradictory results from clinical trials considering the combination of Sorafenib with small molecules in the first line setting were reported [73,74].

## Figures and Tables

**Figure 1 medicina-55-00707-f001:**
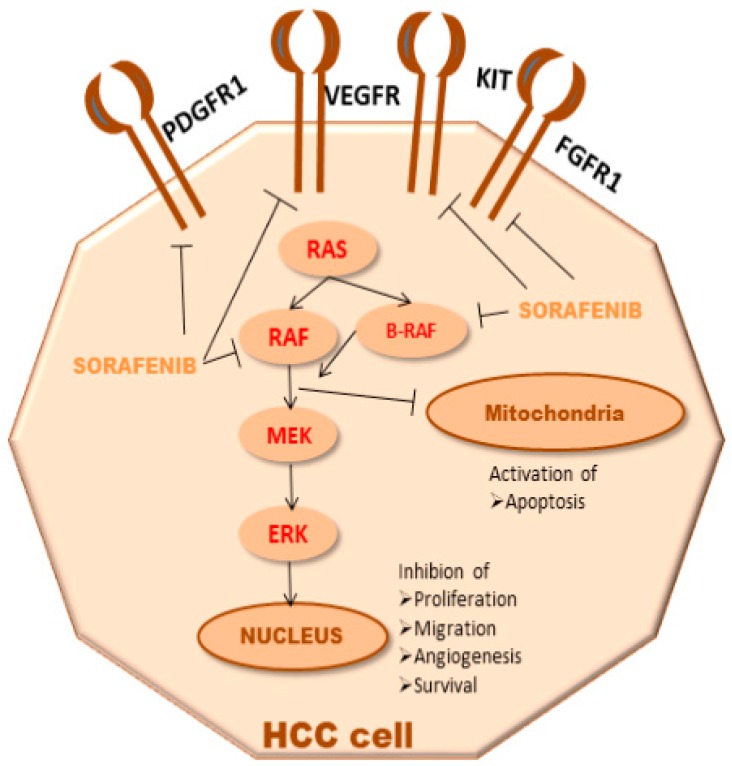
Mechanisms of Sorafenib. A graphic representation of sorafenib mechanisms in HCC patients. Abbreviations—ERK: extracellular signal–regulated kinase; FGFR1: fibroblast growth factor receptors; HCC hepatocellular carcinoma; MEK: mitogen-activated protein kinase; DGFR: platelet-derived growth factor receptors; RAF: VEGF-R: vascular endothelial growth factor receptor.

**Figure 2 medicina-55-00707-f002:**
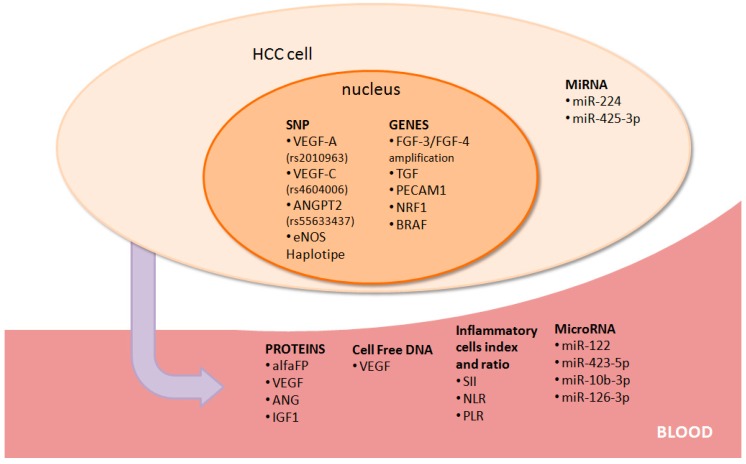
Potential predictive/prognostic markers in hepatocellular carcinoma (HCC) patients treated with sorafenib. A graphic representation of biological prognostic/predictive factors analysed in HCC patients treated with sorafenib.

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
