# Peer review of "Predictive and Prognostic Factors in HCC Patients Treated with Sorafenib"

_medicina, 2019, doi:10.3390/medicina55100707_

Round 1
Reviewer 1 Report
The authors reviewed the data available in the literature with the aim to try to answer the question if it is possible to select patients as candidates for sorafenib treatment, based on clinical or biological predictive and/or prognostic markers. They conclude that it is still long way to go to obtain a panel of clinical and biological biomarkers for the patients who are suitable for sorafenib treatment. As we know, HCC is seriously life threaten, life quality depression and economic burden of society. Sorafenib is a FDA approved drug for unresectable and advanced HCC treatment. The clinical and biological markers for selection of HCC patients to improve treatment response are urgently needed. This study provided the knowledge base for the prognostic prediction of HCC patients with Sorafenib treatment. The reasons for performing such study is fairly straightforward. Here are several comments:
For the clinical markers, need to analyze, in any, the patients’ disease background and family history and treatment history. It would be also help to strengthen the manuscript by adding table(s) for patients information including age, sex, BMI etc. For biological markers, adding subtitle(s) would help to get the point, i.e., Alpha-fetoprotein; Angiogenetic markers, Single Nucleotide Polymorphisms etc. The English issue. It is necessary to have a English-speaking person to further check the language used in this manuscript.
Author Response
For the clinical markers, need to analyze, in any, the patients’ disease background and family history and treatment history.
We tried to review the literature with the aim to find disease background, family history, and treatment history that could be prognostic biomarkers in sorafenib response, anyway, no data are present. So far we added a short conclusion at the end of the paragraph entitled “Clinical predictive/prognostic markers”:
It should be intrigue to discover if disease background, family history, and treatment history that could be prognostic biomarkers in sorafenib response, anyway no data are available in literature concerning these topics.
It would be also help to strengthen the manuscript by adding table(s) for patients information including age, sex, BMI etc.
Being a narrative revision and not a clinical study, it is very difficult to insert clinical-pathological informations of patients of all the studies reviewed. So far, we included a summary table of all the studies.
For biological markers, adding subtitle(s) would help to get the point, i.e., Alpha-fetoprotein; Angiogenetic markers, Single Nucleotide Polymorphisms etc.
We insert subtitles.
The English issue. It is necessary to have a English-speaking person to further check the language used in this manuscript.
A English-speaking person revised the manuscript
Reviewer 2 Report
It is a good review work on the use of Sorafenib
in the treatment of HCC.
I suggest the following
1.- add tables where the information reported so far is summarized. where the study model, finding, mechanism is described
2.- can make some figure where its mechanism of action is described in detail and the routes that are activated or inhibited
3. Sorafenib has an anti-inflammatory or antioxidant or free radical forming effect. Does the nuclear factor Nrf2 participate?
4. What are the perspectives of Sorafenib in the treatment of HCC?
Author Response
1.- add tables where the information reported so far is summarized. where the study model, finding, mechanism is described
We add a single summary table.
2.- can make some figure where its mechanism of action is described in detail and the routes that are activated or inhibited
We added the figure 1, as required.
3. Sorafenib has an anti-inflammatory or antioxidant or free radical forming effect. Does the nuclear factor Nrf2 participate?
We reported data regarding the pivotal role of Nerf2 in HCC
Oxidative stress is a key pathogenic event in the development and progression of HCC (a). nuclear erythroid 2-related factor 2 (Nrf2) is a cytosolic transcription factor regulating the cellular protection via induction of anti-inflammatory, antioxidant, and cyto-protective genes expression (b). In particular, its dysregulation enhances resistance of cancer cells against drugs. So far, the identification of molecules targeting Nrf2 might open new scenario in the prevention and treatment of HCC. Preliminary data in vitro studies showed the ability of Sorafenib to reverse 5-fluorouracil (5-FU) resistance probably through the suppression of Nrf2 expression induced by 5-FU (c), with a putative role in Sorafenib-response prediction.
a) Xu D, Xu M, Jeong S, Qian Y, Wu H, Xia Q, Kong X. The Role of Nrf2 in liver disease: novel molecular mechanisms and therapeutic approaches. Front Pharmacol 2019; 9:1428. doi: 10.3389/fphar.2018.01428
b) Raghunath A, Sundarraj K, Arfuso F, Sethi G, Perumal E. Dysregulation of Nrf2 in hepatocellular carcinoma: role in cancer progression and chemoresistance. Cancers (Basel) 2018; 10(12). pii: E481. doi: 10.3390/cancers10120481
c) Zhou S, Ye W, Duan X, Zhang M, Wang J. The noncytotoxic dose of sorafenib sensitizes Bel-7402/5-FU cells to 5-FU by down-regulating 5-FU-induced Nrf2 expression. Dig Dis Sci 2013; 58(6):1615-26. doi: 10.1007/s10620-012-2537-1
4. What are the perspectives of Sorafenib in the treatment of HCC?
We cited data concerning “perspectives of Sorafenin in the treatment of HCC” in the Conclusion section
Furthermore, in vitro and in vivo preclinical models evaluating the combination of Sorafenib with drugs able to overcome its acquired resistance mechanisms have been performed (a). Indeed, contradictory results from clinical trials considering the combination of Sorafenib with small molecules in the first line setting have been reported (b, c).
a) Gavini J, Dommann N, Jakob MO, Keogh A, Bouchez LC, Karkampouna S, Julio MK, Medova M, Zimmer Y, Schläfli AM, Tschan MP, Candinas D, Stroka D, Banz V. Verteporfin-induced lysosomal compartment dysregulation potentiates the effect of sorafenib in hepatocellular carcinoma. Cell Death Dis. 2019 10(10):749. doi: 10.1038/s41419-019-1989-z.
b) Kelley RK, Gane E, Assenat E, Siebler J, Galle PR, Merle P, Hourmand IO, Cleverly A, Zhao Y, Gueorguieva I, Lahn M, Faivre S, Benhadji KA, Giannelli G. A phase 2 study of Galunisertib (TGF-β1 Receptor Type I Inhibitor) and Sorafenib in patients with advanced hepatocellular carcinoma. Clin Transl Gastroenterol. 2019 10(7):e00056. doi: 10.14309/ctg.0000000000000056.
c) Abou-Alfa GK, Miksad RA, Tejani MA, Williamson S, Gutierrez ME, Olowokure OO, Sharma MR, El Dika I, Sherman ML, Pandya SS. A phase Ib, open-label study of Dalantercept, an Activin Receptor-Like Kinase 1 Ligand Trap, plus Sorafenib in advanced hepatocellular carcinoma. Oncologist. 2019 24(2):161-e70. doi: 10.1634/theoncologist.2018-0654. Epub 2018 Oct 23.